# Anti-Adapter Armor: A Universal and Authentication-Integrated Framework for Preventing Unauthorized Zero-Shot Image-to-Image Generation

## Abstract

With the advancement of diffusion models, image generation has entered an era of zero-shot image-to-image synthesis, where highly similar facial identities or artistic styles can be produced using only a single portrait or artwork as input, without requiring any model parameter fine-tuning. However, while these technologies offer significant benefits to artistic creation, they simultaneously introduce non-negligible risks associated with right infringement, such as the unauthorized forgery of facial identities and the plagiarism of artistic styles. To address these risks, this paper proposes **Anti-Adapter Armor**, the first **universal** and **authentication-integrated** framework designed to protect personal images against unauthorized zero-shot image-to-image generation. We begin by analyzing how existing zero-shot image-to-image methods utilize image encoders to convert input images into embeddings, which are injected into the diffusion model's UNet via cross-attention. Based on this, we develop a reversible encryption framework that transforms original image embeddings into diverse encrypted forms based on different passwords. Authorized users can recover the original embeddings using the decryptor and correct passwords for normal image generation. To achieve protection, we propose a multi-targeted adversarial attack that transfers the original image embeddings into the encrypted forms by adding adversarial perturbation. Therefore, the protected images are equipped with a protective coating that restricts unauthorized users to generating encrypted content exclusively. Extensive experiments show that our approach outperforms state-of-the-art protection methods in preventing unauthorized zero-shot image-to-image generation, while enabling adaptable and secure authentication for authorized users.

## 1 Introduction

With the development of diffusion models, image generation has evolved from early text-to-image methods Rombach et al. (2022) to the current framework that supports multimodal conditional inputs, in which text and visual inputs jointly guide the generation process. Recently, many zero-shot image-to-image generation approaches are proposed, which can synthesize highly similar facial identities or artistic styles with only a single portrait or artwork as input. Unlike fine-tuning diffusion models, these approaches do not alter the pre-trained parameters of diffusion models. Instead, they inject the key information of reference images into diffusion models in a plug-and-play manner, thereby offering substantial convenience to AI-generated content (AIGC) creators. However, despite their widespread adoption, these technologies have also introduced security risks associated with image contents. The primary concern lies in copyright infringement: unauthorized synthesis of facial identity and artistic style may violate individuals' rights to their portrait and artists' intellectual property rights. Moreover, the generation of sensitive or harmful content using personal portraits could lead to adverse societal consequences. For this concern, although deepfake detection technology has made significant progress, it remains reactive, addressing problems only after they occur. Therefore, to protect personal images from unauthorized zero-shot image-to-image generation, there is an urgent need to develop a proactive protection mechanism that establishes AIGC usage permissions at the source of image generation.

Compared to fine-tuning diffusion models, the zero-shot image-to-image generation approaches treat images in a similar way as text prompts. These approaches use an image encoder to extract image embeddings and incorporate an additional cross-attention module that decouples the embeddings of text and image prompts. In practical scenarios, such methods can replicate the target facial identity or artistic style using only a single image as reference. Compared to fine-tuning-based approaches, these zero-shot methods present a more significant risk, as unauthorized users can achieve their goals with just one image, which is considerably easier to obtain than the larger dataset required for fine-tuning. Therefore, preventing unauthorized generation with zero-shot image-to-image methods requires increased attention, as no universal and flexible solution has yet been developed to address this challenge. Recently,

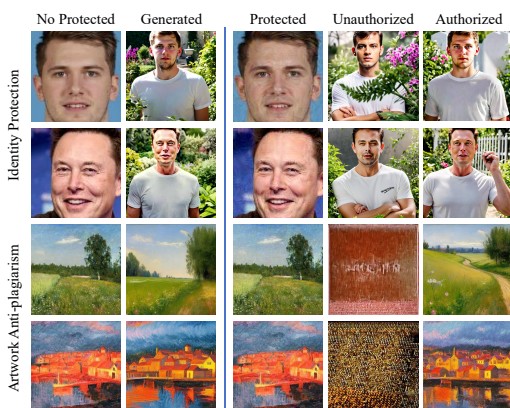

Figure 1: When publishing the original image without protection, diffusion models can easily imitate the content via just one image (**the second column**). After applying Anti-Adapter Armor (**the third column**), unauthorized users cannot generate the target content (**the fourth column**) while authorized users can generate intended images through a password-based authentication (**the fifth column**).

Song *et al.* Song et al. (2025) propose IDProtector, a data poisoning-based method tailored to facial identity protection. However, IDProtector has two significant limitations: (1) this method is irreversible, and even trusted parties cannot recover the true identity, thereby limiting its flexibility; (2) it focuses exclusively on unauthorized identity forgery while overlooking the other critical issue in real scenarios: style plagiarism of artworks. Additionally, this method is closed-source.

To safeguard personal images from being forged or plagiarized by unauthorized zero-shot image-to-image generation methods, this paper begins by systematically analyzing the major challenges in this scenario. There are two primary challenges which remain insufficiently addressed: **(1) Universality:** The protection solution should be effective across various zero-shot image-to-image methods and diverse threat scenarios such as identity forgery and artistic style plagiarism. **(2) Authentication:** The image owner holds the authority to define permitted usage scenarios. While image post-processing operations can remove added protection information from protected images, unauthorized users may also exploit this technique. Therefore, the protection solution must exhibit robustness against common post-processing techniques while permitting authorized users to produce intended outputs securely.

To address these challenges, this paper proposes **Anti-Adapter Armor**, the first **universal** and **authentication-integrated** framework against unauthorized zero-shot image-to-image generation. As shown in Fig. 2, the framework first encrypts the original image embeddings required by zero-shot image-to-image generation methods. Then, a protective coating related to the encrypted embeddings are added to the original image through the proposed multi-targeted adversarial attack. Consequently, unauthorized users cannot replicate the original content based on the protected image, while authorized users can recover the original embeddings for intended generation using the correct decryption password. The protection results are shown in Figure 1. The main contributions of this work include:

- We propose the first **universal** and **authentication-integrated** framework for preventing unauthorized zero-shot image-to-image generation.

- The proposed framework provides flexible access control through password-based authentication, allowing different passwords to generate diverse protected images.

- The proposed framework shows superior universality across various tasks and diverse fine-tuning methods.

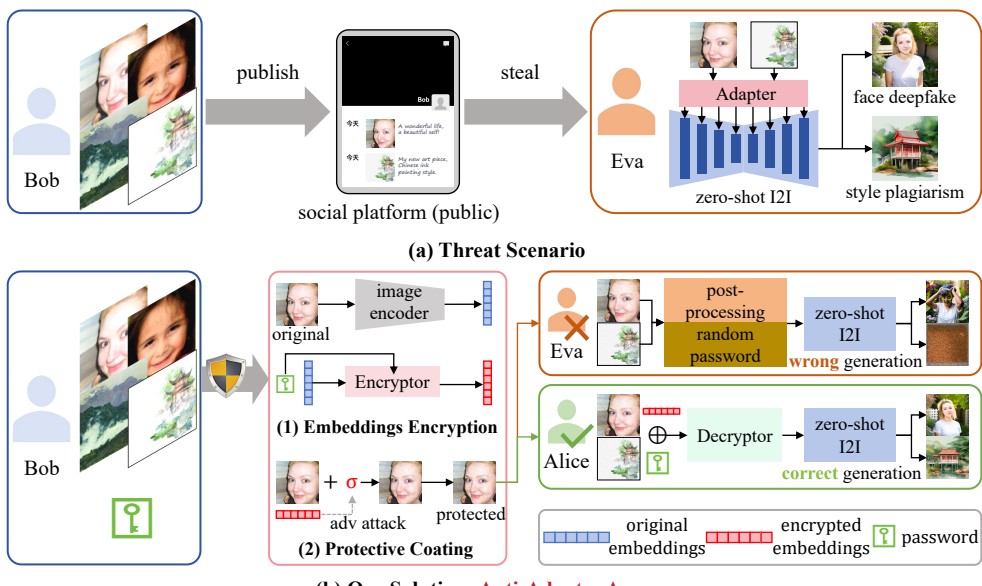

Figure 2: (a) the threat scenario of this paper: personal images can be imitated by zero-shot image-to-image generation based on diffusion models. (b) the proposed solution (the red box): adding a protective coating to the original images, ensuring **universal** and **authentication-integrated**.

## 2 RELATED WORK

### 2.1 ZERO-SHOT IMAGE-TO-IMAGE GENERATION BASED ON DIFFUSION

Fine-tuning technologies for diffusion models, which involve optimizing all or part of the model parameters using a small-scale dataset, enable the generation of content that closely resembles the images used during fine-tuning. However, these methods still require multiple images representing a specific character or style, such as LoRA Hu et al. (2022), DreamBooth Ruiz et al. (2023), Textual Inversion Gal et al. (2022), and Custom Diffusion Kumari et al. (2023). Nevertheless, acquiring such multiple images is often challenging in practice. To address this, zero-shot image-to-image generation methods have been developed, requiring only a single image to produce similar content. These methods use an image encoder to extract embeddings from the single reference image and a cross-attention module to integrate them into specific UNet layers. For general generation, IP-Adapter Ye et al. (2023) employs CLIP Radford et al. (2021) as the image encoder. For facial identity generation, IP-Adapter Faceid Ye et al. (2023) and Instant-ID Wang et al. (2024) encode embeddings through pretrained ArcFace Deng et al. (2019) models. Recent methods such as Photomakerr Li et al. (2024), PULID Guo et al. (2024), and StoryMaker Zhou et al. (2024) further integrate both CLIP and ArcFace encoders to enhance identity preservation. Compared to fine-tuning, these zero-shot methods alleviate the necessity of acquiring multiple fine-tuning images, thereby enhancing their practical applicability.

#### 2.1.1 IMAGE PROTECTION FOR DIFFUSION MODELS

The growing use of fine-tuning technologies in diffusion models has raised increasing concerns about the unauthorized use of personal images. To address this risk, numerous methods have been proposed to protect copyrighted content, such as artistic styles Shan et al. (2023) and facial identities Van Le et al. (2023), from being reproduced by fine-tuning. Adv-DM (mist) Liang et al. (2023) targets the fine-tuning of diffusion models to output a predefined noisy image by adding pixel-level adversarial perturbations to original images. CAAT Xu et al. (2024) demonstrates that subtle perturbations in the attention mechanism can induce strong fine-tuning misdirection. Pretender Sun et al. (2025) proposes an adversarial training framework to effectively mislead downstream fine-tuning processes, demonstrating universality across various fine-tuning methods. ACE Zheng et al. (2023) introduce a unified target to guide perturbation optimization consistently across both the forward

encoding and reverse generation processes, effectively addressing the offset problem in this field and enhancing protection stability and transferability. Nightshade Shan et al. (2024) implements a prompt-specific poisoning approach to mislead text-to-image models to generate incorrect results. For anti-plagiarism of artistic style, Glaze Shan et al. (2023) proposes an optimization-based cloaking algorithm to obstruct the learning and replication of artistic style by diffusion models. For defending specific fine-tuning methods, Anti-DreamBooth Van Le et al. (2023) embeds imperceptible noise into training images to prevent fine-tuning by DreamBooth. In preventing zero-shot image-to-image generation, IDProtector Song et al. (2025) introduces a unified model that adds adversarial perturbations to disrupt the image encoders used by such methods.

## 3 METHODOLOGY

### 3.1 THREAT SCENARIO

This section presents the real-world threat scenario faced by sharing personal images. We define three key parties involved: **(1) Bob**, **(2) Alice**, and **(3) Eva**. The specific objectives of each party are outlined as follows:

- **Bob:** as shown in Figure 2, Bob is the images' owner who hopes to publicly share his personal portraits or paintings. For the authorized party Alice, Bob permits the use of zero-shot image-to-image generative models to create derivative works from his images. In contrast, the unauthorized party Eva is prohibited from using such models to replicate his images.

- **Alice:** the authorized party with rights to use Bob's images. Given the correct password, Alice can recover the original image embeddings, enabling normal usage.

- **Eva:** the unauthorized adversary attempting illicit usage. As shown in Figure 2 (b), Eva may try to remove the protection solution by applying post-processing operations to Bob's images. Furthermore, in a more dangerous scenario, Eva obtains the decryption tool and attempts to guess the password to recover the original embeddings.

### 3.2 PIPELINE OF ANTI-ADAPTER ARMOR

Based on the above analysis, existing zero-shot image-to-image generation methods generally employ an image encoder to extract image embeddings, and incorporate an additional cross-attention module to decouple the projected image embeddings from the text prompt embeddings. Therefore, the core idea of our solution is to disrupt the original embeddings of images, making them deviate from the initial form. To achieve this objective, Anti-Adapter Armor consists of two sequential stages: **(1) embeddings encryption** and **(2) protective coating generation**. We define the image encoder as **IE**, the diffusion model as **DM**, the proposed encryptor as **Enc**, the password as $\mathcal{P}_{crt}$, the original image as $\mathcal{I}_{ori}$, and the image embeddings as $\mathcal{E}$. The password and the embeddings to be encrypted have the same dimension.

#### 3.2.1 STAGE-1: EMBEDDINGS ENCRYPTION

**Overall Pipeline:** The image encoder **IE** is first used to extract original image embeddings $\mathcal{E}_{ori}$. Subsequently, $\mathcal{E}_{ori}$ are encrypted into $\mathcal{E}_{enc}$ by encryptor **Enc**, with the requirement that the similarity between $\mathcal{E}_{enc}$ and $\mathcal{E}_{ori}$ be as low as possible. This process can be formulated as:

$$\mathcal{E}_{enc} = \mathbf{Enc}(\mathcal{E}_{ori}, \mathcal{P}_{crt}), \tag{1}$$

where $\mathcal{E}_{ori} = \mathbf{IE}(\mathcal{I}_{ori})$. For the authorized user Alice, the encrypted embeddings $\mathcal{E}_{enc}$ can be decrypted into $\mathcal{E}_{dec}$ using the decryptor **Dec** and the correct password $\mathcal{P}_{crt}$. This process achieves **authentication** for authorized users, which can be formulated as:

$$\mathcal{E}_{dec} = \mathbf{Dec}(\mathcal{E}_{enc}, \mathcal{P}_{crt}). \tag{2}$$

**Architecture of Encryptor and Decryptor:** As illustrated in Figure 3, both the encryptor **Enc** and decryptor **Dec** are trainable models sharing the same architecture but differ in their parameters. The proposed encryptor and decryptor are composed of a self-attention module, a cross-attention

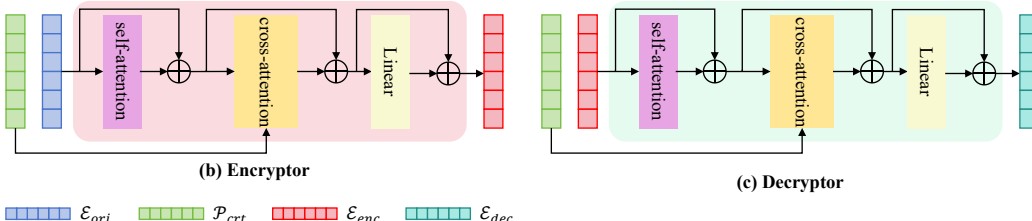

(b) Encryptor

(c) Decryptor

$\mathcal{E}_{ori}$    $\mathcal{P}_{crt}$    $\mathcal{E}_{enc}$    $\mathcal{E}_{dec}$

Figure 3: The details of the encryptor and decryptor. We omit layer normalization operation here.

module, and a fully connected layer. The cross-attention module enables the encrypted results to be dependent on the corresponding passwords.

**Optimization Objectives:** The primary optimization objectives involve minimizing the similarity between encrypted embeddings $\mathcal{E}_{enc}$ and original embeddings $\mathcal{E}_{ori}$, while simultaneously maximizing the similarity between decrypted embeddings $\mathcal{E}_{dec}$ and the original embeddings $\mathcal{E}_{ori}$. To achieve these objectives, we propose an encryption loss $\mathcal{L}_{enc}$ and a decryption loss $\mathcal{L}_{dec}$ based on cosine similarity **CosSim** inspired by facial de-identification Li et al. (2023); Gu et al. (2020); Cao et al. (2021), which guide the joint training of the encryptor and decryptor. When training, we randomly generate $n{+}1$ passwords that includes one correct password $\mathcal{P}_{crt}$ and $n$ wrong password $\mathcal{P}_{wrg\_i}$, $i{=}\{0, 1, ..., n{-}1\}$. The loss functions are formulated as follows:

$$\mathcal{L}_{enc} = \textbf{CosSim}(\textbf{Enc}(\mathcal{E}_{ori}, \mathcal{P}_{crt}), \mathcal{E}_{ori}) + \sum_{i=0}^{n-1} \textbf{CosSim}(\textbf{Enc}(\mathcal{E}_{ori}, \mathcal{P}_{wrg\_i}), \mathcal{E}_{ori}), \quad (3)$$

$$\mathcal{L}_{dec} = 1 - \textbf{CosSim}(\textbf{Dec}(\mathcal{E}_{enc\_crt}, \mathcal{P}_{crt}), \mathcal{E}_{ori}), \quad (4)$$

where $\mathcal{E}_{enc\_crt}{=}\textbf{Enc}(\mathcal{E}_{ori}, \mathcal{P}_{crt})$. The range of cosine similarity is restricted between 0 and 1 to ensure the non-negativity. Furthermore, to prevent unauthorized users from recovering the original embeddings using random passwords, we propose $\mathcal{L}_{wrg}$ as follows:

$$\mathcal{L}_{wrg} = \sum_{i=0}^{n-1} \textbf{CosSim}(\textbf{Dec}(\mathcal{E}_{enc\_crt}, \mathcal{P}_{wrg\_i}), \mathcal{E}_{ori}). \quad (5)$$

To enhance the diversity of encryption and decryption results, we propose $\mathcal{L}_{div}$. For each iterative optimization, the batch size is $b$. Thus, there are a total of $n{+}1$ encrypted embeddings and $n$ wrong decrypted embeddings in each iteration. We re-number the above $b \times (2n{+}1)$ embeddings as from 0 to $N$, where $N{=}b \times (2n{+}1){-}1$. $\mathcal{L}_{div}$ is formulated as follows:

$$\mathcal{L}_{div} = \frac{1}{2} \sum_{k=0}^{N} \sum_{j=0}^{N} \textbf{CosSim}(\mathcal{E}_k, \mathcal{E}_j), \ s.t. k \neq j. \quad (6)$$

This loss function enforces diverse encryption and decryption results when different passwords are applied, while ensuring that distinct original embeddings do not generate similar encrypted or decrypted results with different passwords.

Finally, to prevent similarity in encryption and decryption results when the same password is applied to different original embeddings, we introduce $L_{div\_s}$. Specifically, in each iteration, for each original embedding in the same batch, two fixed passwords, $\mathcal{P}_{enc}$ and $\mathcal{P}_{dec}$, are respectively used for encryption and decryption. $\mathcal{L}_{div\_s}$ is formulated as follows:

$$\mathcal{L}_{div\_s} = \frac{1}{2} \sum_{k=0}^{b-1} \sum_{j=0}^{b-1} \textbf{CosSim}(\mathcal{E}_{enck}, \mathcal{E}_{encj}) + \frac{1}{2} \sum_{k=0}^{b-1} \sum_{j=0}^{b-1} \textbf{CosSim}(\mathcal{E}_{deck}, \mathcal{E}_{decj}) \ s.t. k \neq j, \quad (7)$$

where $\mathcal{E}_{enc}$ and $\mathcal{E}_{dec}$ represent the encrypted embeddings with $\mathcal{P}_{enc}$ and the decrypted embeddings with $\mathcal{P}_{dec}$, respectively. In conclusion, the total loss function is the sum of the above-mentioned loss functions weighted by $\lambda_i$:

$$\mathcal{L} = \lambda_1 \mathcal{L}_{enc} + \lambda_2 \mathcal{L}_{dec} + \lambda_3 \mathcal{L}_{wrg} + \lambda_4 \mathcal{L}_{div} + \lambda_5 \mathcal{L}_{div\_s}. \quad (8)$$

Table 1: Comparison results of facial identity protection and artwork anti-plagiarism tasks against SOTA methods. ISM: identity similarity between generated and original faces, AFR: abnormal face rate, ESM: embedding similarity between protected and original artworks. "specific" refers to the consideration of only a single type of image encoder, whereas "general" considers all types of image encoders. The best and second-best results are marked by red and blue.

| Method | Face Identity Protection | | | | | | Artwork Anti-Plagiarism | | | | | |
| | IP-Adapter Faceid | | | Instant-ID | | | IP-Adapter | | | IP-Adapter Plus | | |
| | ISM↓ | AFR↑ | PSNR↑ | ISM↓ | AFR↑ | PSNR↑ | ESM↓ | PSNR↑ | LPIPS↓ | ESM↓ | PSNR↑ | LPIPS↓ |
| No Protect | 1.0 | 0.00 | NA | 1.0 | 0.00 | NA | 1.0 | NA | NA | 1.0 | NA | NA |
| Pretender | 0.8691 | 0.01 | 30.08 | 0.8721 | 0.02 | 30.08 | 0.8164 | 30.09 | 0.1307 | 0.7393 | 30.09 | 0.1307 |
| Adv-DM | 0.8975 | 0.01 | 29.43 | 0.9092 | 0.01 | 29.43 | 0.8438 | 28.05 | 0.1488 | 0.7538 | 28.05 | 0.1488 |
| ACE | 0.9302 | 0.04 | 27.00 | 0.9346 | 0.02 | 27.00 | 0.7920 | 26.71 | 0.2042 | 0.7244 | 26.71 | 0.2042 |
| CAAT | 0.9561 | 0.05 | 32.08 | 0.9600 | 0.02 | 32.08 | 0.8467 | 32.55 | 0.1260 | 0.7326 | 32.55 | 0.1260 |
| ours(specific) | 0.0514 | 0.15 | 30.01 | -0.011 | 0.16 | 30.26 | 0.0161 | 30.62 | 0.1109 | 0.2390 | 29.01 | 0.1505 |
| ours(general) | 0.1422 | 0.07 | 32.10 | 0.0685 | 0.19 | 32.10 | 0.1175 | 30.81 | 0.0980 | 0.2713 | 30.81 | 0.0980 |

### 3.2.2 STAGE-2: PROTECTIVE COATING GENERATION

**Overall Objectives:** In the second stage, we generate a protective coating on the original image $\mathcal{I}_{ori}$ by optimizing imperceptible perturbations $\delta$. The adversarial perturbation budget is denoted by $\epsilon$. The generation process of protected image $\mathcal{I}_{pro}$ can be formulated as:

$$
\begin{aligned}
\mathcal{I}_{pro} &= \mathcal{I}_{ori} + \delta \\
s.t. \ &\max(\textbf{CosSim}(\textbf{IE}_i(\mathcal{I}_{pro}), \mathcal{E}_{tar\_i})) \\
&\text{and } |\delta| \leq \epsilon, i = \{0, 1, ..., m-1\},
\end{aligned}
\tag{9}
$$

where $\mathcal{E}_{tar\_i}$ denotes the encrypted embedding associated with the image encoder employed by the $i^{th}$ zero-shot image-to-image method. There are a total of $m$ encoders.

**Robust Multi-targeted Adversarial Attack:** As shown in Eq. 9, the objective of protective coating generation is to modify the original image embeddings $\mathcal{E}_{ori}$ to the encrypted embeddings $\mathcal{E}_{tar\_i}$ without significantly altering the original image quality. This process faces two primary challenges: (1) unauthorized adversaries may employ more than one zero-shot image-to-image generation methods, necessitating that the protective coating exhibits universality across various image encoders utilized in these methods; (2) unauthorized adversaries may attempt to erase the protective coating through image processing operations, such as blurring and noise addition. To effectively tackle these challenges, this paper proposes a robust multi-targeted adversarial attack method based on Fast Gradient Sign Method (FGSM).

To achieve **universality**, we design a multi-targeted adversarial loss function $\mathcal{L}_{mt}$ as follows:

$$
\mathcal{L}_{mt} = \sum_{i=0}^{m-1} 1 - \textbf{CosSim}(\mathcal{E}_{pro\_i}, \mathcal{E}_{tar\_i}),
\tag{10}
$$

where $\mathcal{E}_{tar\_i} = \textbf{Enc}_i(\textbf{IE}_i(\mathcal{I}_{ori}), \mathcal{P})$ and $\mathcal{E}_{pro\_i} = \textbf{IE}_i(\mathcal{I}_{pro})$. Then, the perturbations $\delta$ are updated by the gradients $\nabla$ of $\mathcal{I}_{ori}$:

$$
\delta = \delta - \sigma * \nabla_{\mathcal{I}_{adv}} \mathcal{L}_{mt}.
\tag{11}
$$

The range of $\delta$ is restricted between $-\epsilon$ and $+\epsilon$. To enhance robustness, we add differentiable image processing operations **diff_distortion** in each iteration. The above procedure is repeated until the similarity surpasses the predefined threshold $ths$. This algorithm is detailed in Appendix.

## 4 EXPERIMENTS

### 4.1 EXPERIMENTAL SETTINGS

**Dataset:** We evaluate the proposed approach on two typical tasks: facial identity protection and artwork anti-plagiarism. For facial identity protection, we use CelebA Liu et al. (2015) for training and 200 facial images from FFHQ Karras et al. (2019) for testing. All images are normalized to $112 \times 112$ by face alignment. For artwork anti-plagiarism, we select 25,769 painting images from Wikiart Saleh & Elgammal (2015) as the training dataset and 50 painting images unseen in the training phase for test.

| NP | P (clean) | P (noise) | P (blur) | P (JPEG) | | NP | P (clean) | P (noise) | P (blur) | P (JPEG) |

Figure 5: Robustness results of face identity protection and artwork anti-plagiarism.

**Implementation Details:** In experiments, we select two distinct zero-shot image-to-image generation methods for each task ($m$=2 in Eq. 9). In the case of facial identity protection, IP-Adapter FaceID Ye et al. (2023) based on SD-1.5 and Instant-ID Wang et al. (2024) based on SDXL are selected. These methods use the same image encoder (ArcFace Deng et al. (2019)) but differ in terms of their pre-trained models. For artwork anti-plagiarism, we choose IP-Adapter Ye et al. (2023) and IP-Adapter-Plus Ye et al. (2023), which utilize different layers of the embeddings encoded by CLIP Radford et al. (2021). We train a pair of encryptor and decryptor for each image encoder. For the hyperparameter settings, we set the loss weights $\lambda_1$ to 1, $\lambda_2$ to 5, $\lambda_3$ to 1, $\lambda_4$ to 1, and $\lambda_5$ to 1. The similarity threshold $ths$ is set to 0.75 and 0.65 for two tasks, respectively. The budget $\epsilon$ is set to $\frac{11}{255}$ and $\frac{21}{255}$, respectively. All experiments are conducted on two NVIDIA 4090 GPUs.

## 4.2 COMPARATIVE RESULTS

We compare Anti-Adapter Armor with four state-of-the-art open source methods: Pretender Sun et al. (2025), Adv-DM Liang et al. (2023), ACE Zheng et al. (2023), and CAAT Xu et al. (2024). These methods are tailored to protect images from being utilized to fine-tune diffusion models. For facial identity protection, we employ the identity cosine similarity (ISM) and abnormal face rate (such as occlusion and abnormal patterns in Figure 6) of generated images (AFR) as metrics. A lower ISM value combined with a higher AFR value indicates a more effective protection performance. For artwork anti-plagiarism, the embeddings cosine similarity (ESM) is utilized to assess the protection effectiveness. In addition, we exploit LPIPS Zhang et al. (2018) to evaluate the visual quality of protected images. For fair comparison, the PSNR of

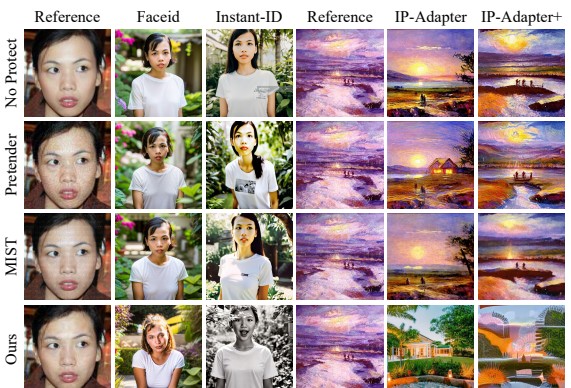

Figure 4: Visualization results of comparative experiments. "**Reference**" denotes the image prompt provided as input to FaceID/Instant-ID/IP-Adapter/IP-Adapter Plus. "**No Protect**" represents the original images without any protection.

protected images across these methods are kept around 30. As shown in Table 1, our approach demonstrates universality across various zero-shot image-to-image generation methods and tasks, thereby safeguarding images against unauthorized generation. In contrast, existing methods are tailored to defend fine-tuning diffusion models, exhibiting limited generalization capacity to zero-shot methods. We present visualization results in Figure 4.

## 4.3 ABLATION STUDY

As shown in Table 2, the ablation experiments are divided into two parts. The diversity is defined as the cosine similarity of two embeddings.

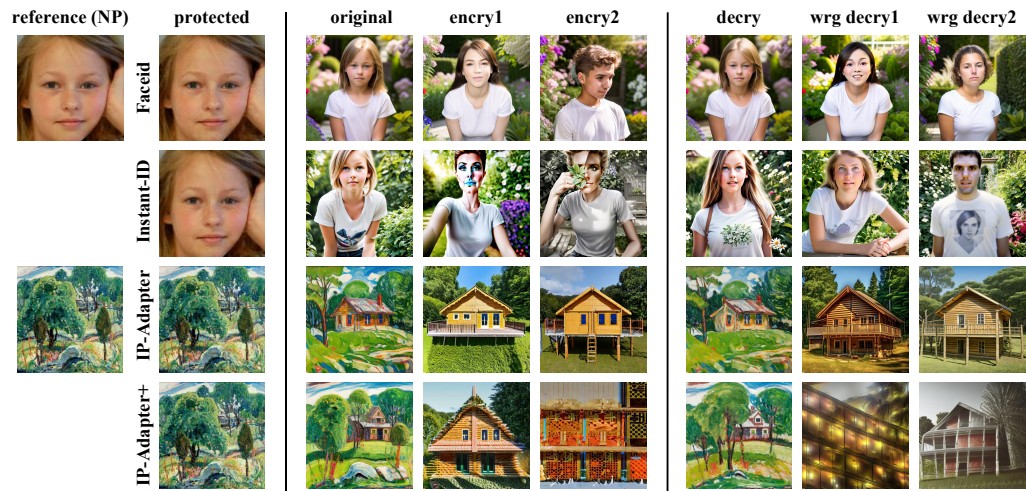

Figure 6: The visualization results of encryption and decryption performance. More visualization results are presented in Appendix. Text prompts: "*a young woman in white T-shirt in a garden*" and "*best quality, high quality, a wooden house in forest*".

Table 2: Ablation study of different components. "Diversity Same pwd" represents the cosine similarity of embeddings encrypted by the same password for different images. "Wrong Dec Rate" represents the success rate of recovering original embeddings by random passwords. The cosine similarity threshold for successful recovery is set to 0.8.

| Sub-Table | Model | Facial Identity Protection | | | | Artwork Anti-Plagiarism | | | |
|---|---|---|---|---|---|---|---|---|---|
| | | Encrypt Diversity↓ | Decrypt Diversity↓ | Diversity Same pwd↓ | Wrong Dec Rate↓ | Encrypt Diversity↓ | Decrypt Diversity↓ | Diversity Same pwd↓ | Wrong Dec Rate↓ |
| Part 1 | w/o $\mathcal{L}_{div},\mathcal{L}_{div\_s}$ | 0.9849 | 0.9995 | 0.9993 | **0.0** | 0.4563 | 0.9927 | 0.9160 | **0.0** |
| | w/o $\mathcal{L}_{div\_s}$ | **-0.0137** | **0.0100** | 0.9958 | **0.0** | **0.0253** | **0.0159** | 0.9788 | **0.0** |
| | with $\mathcal{L}_{div},\mathcal{L}_{div\_s}$ | 0.0232 | 0.0468 | **0.0361** | 0.015 | 0.082 | 0.086 | **0.5891** | 0.05 |
| | Method | Specific Similarity↓ | Unseen Similarity↓ | PSNR↑ | Time(s)↓ | Specific Similarity↓ | Unseen Similarity↓ | PSNR↑ | Time(s) ↓ |
| Part 2 | w/o $\mathcal{L}_{mt}$ | **0.02** | 0.2946 | 30.14 | **25.13** | **0.1276** | 0.5128 | 29.82 | **220.90** |
| | with $\mathcal{L}_{mt}$ | 0.1054 | **0.1054** | **32.10** | 40.03 | 0.1944 | **0.1944** | **30.81** | 480.55 |

**Part 1: Encryption and Decryption.** In this part, we evaluate the effect of $\mathcal{L}_{div}$ and $\mathcal{L}_{div\_s}$. The first sub-table in Table 2 demonstrates that $\mathcal{L}_{div}$ can substantially enhance the diversity of the encrypted and decrypted results across various passwords. Furthermore, with the help of $\mathcal{L}_{div\_s}$, the results conditioned on the same password can also exhibit substantial diversity as measured by **"Diversity Same pwd"**. We also evaluate the security of our approach by wrong decryption rate which represents the successful rate of decrypting with random passwords. Due to the balance requirement between diversity and encryption/decryption performance, the introduction of $\mathcal{L}_{div}$ and $\mathcal{L}_{div\_s}$ slightly increases the security risk of random password attack. The maximum wrong decryption rate remains as low as 5%, which has slight impact on the overall security of the proposed method.

**Part 2: the Impact of $\mathcal{L}_{mt}$.** Finally, we conduct ablation experiments for $\mathcal{L}_{mt}$ in the proposed multi-targeted adversarial attack. In the third sub-Table of Table 2, **"w/o $\mathcal{L}_{mt}$"** represents $m = 1$ in Eq. 9 and **"with $\mathcal{L}_{mt}$"** represents $m = 2$ in Eq. 9. The metric **"Specific Similarity"** is the cosine similarity of embeddings extracted by the utilized image encoder which is different in the case of **"w/o $\mathcal{L}_{mt}$"**. The metric **"Unseen Similarity"** corresponds to the cosine similarity of embeddings extracted by other image encoders which is not considered in the case of **"w/o $\mathcal{L}_{mt}$"**. The lower the values of these two metrics, the greater the distinction between the protected images and their original forms. These results demonstrate that $\mathcal{L}_{mt}$ can enhance the generalization capacity of protected images against various zero-shot image-to-image generation methods.

## 4.4 ROBUSTNESS EVALUATION

This section evaluates the robustness against potential posting-processing operations applied by unauthorized users. We consider three categories of common image distortions that cannot im-

pair generation quality: Gaussian noise, Gaussian blur, and JPEG compression. In the experiments of robustness evaluation, we select Gaussian noise, Gaussian blur, and JPEG compression as the post-processing operations. The mean and standard deviation of Gaussian noise are 0 and 0.01, respectively. The kernel size and standard deviation of Gaussian blur are $3 \times 3$ and $0.4$. The quality factor of JPEG compression is 0.9.

The above settings are justified by two key factors: (1) these operations are common image processing techniques that do not require the execution of complex programs and can be easily performed using smartphone or user-friendly software. (2) serious distortion may compromise the inherent content of images, resulting in outputs that are blurred or contain artifacts, which is undesirable for unauthorized users. The robustness results presented in Table 3 indicate that our protection solution is resilient enough to Gaussian noise and Gaussian blur because we augment the

Table 3: Robustness evaluation under different distortions. The results denote the cosine similarity of embeddings between original images and distorted protected images. "Clean" represents protected images without distortions.

| Type | Faceid | Ins-ID | IP-Ada | IP-Ada+ |
|------|--------|--------|--------|---------|
| Clean | 0.0514 | -0.0110 | 0.0107 | 0.2523 |
| Noise | 0.1678 | 0.1742 | 0.1852 | 0.3864 |
| Blur | 0.1407 | 0.1285 | 0.0553 | 0.2890 |
| JPEG | 0.3682 | 0.3882 | 0.6870 | 0.6675 |

pipeline of multi-targeted adversarial attack by adding differentiable image processing operations in iterations. For JPEG compression, the robustness of our approach is lower compared to the other two types of distortions, which suggests that the introduced adversarial perturbations are fragile to discrete cosine transform and quantization operations. We present some visualization results in Figure 5.

## 4.5 Performance of Encryption and Decryption

As the first authentication-integrated framework in this field, our approach can decrypt the encrypted embeddings to their original form with a correct password. As described in optimization objectives, the primary objective of the encryptor is to maximize the similarity between the encrypted embeddings and the original embeddings, whereas the decryptor aims to recover the encrypted embeddings that are as close as possible to the original ones. We evaluate the effectiveness of encryptor and decryptor through the cosine similarity be-

Table 4: Effects of encryption and decryption. Diversity is evaluated by computing the cosine similarity of embeddings encrypted or decrypted by different passwords. Lower similarity represents better diversity.

| Model | Encrypt Effect↓ | Decrypt Effect↑ | Encrypt Diversity↓ | Decrypt Diversity↓ |
|-------|--------|--------|--------|--------|
| Faceid | -0.0320 | 0.9927 | 0.0203 | 0.0434 |
| Ins-ID | -0.0544 | 0.9725 | 0.0591 | 0.0966 |
| IP-Ada. | -0.0244 | 0.9312 | 0.0509 | 0.1230 |
| IP-Ada+ | -0.0067 | 0.9217 | 0.1120 | 0.0612 |

tween the encrypted/decrypted embeddings and the original embeddings. For diversity, we define encryption and decryption diversity as the cosine similarity between embeddings generated from the same input following encryption and decryption processes using different random passwords. Table 4 indicates that our approach achieves good performance on encryption effect, decryption effect, encryption diversity, and decryption diversity. We present visualization results in Figure 6.

## 5 Conclusion

This paper introduces Anti-Adapter Armor, the first universal and authentication-integrated framework designed to prevent unauthorized zero-shot image-to-image generation. Experiments show our method effectively defends against various zero-shot generation techniques across tasks, proving its broad applicability. It also allows authorized users to recover original embeddings using password-based authentication. Future work will focus on improving robustness and visual quality.

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

## A    APPENDIX

### A.1    MORE IMPLEMENTATION DETAILS

For facial identity protection, Faceid uses the buffalol-pretrained model of ArcFace Deng et al. (2019) and Instant-ID uses the antelopev2-pretrained model of ArcFace. The embedding dimensionality of ArcFace is $1 \times 512$. For artwork anti-plagiarism, IP-Adapter uses the embedding encoded by CLIP, which is a $1 \times 1024$ tensors. IP-Adapter Plus uses the hidden state of CLIP Radford et al. (2021), which is a $1280 \times 768$. To save computing resources, we randomly sample $b$ row vectors from the 1280 row vectors corresponding to one image for one iteration. In the experiments, $b$ is set to 32 and 8 for the two tasks, respectively.

### A.2    ALGORITHM DETAILS OF ROBUST MULTI-TARGETED ADVERSARIAL ATTACK

### A.3    MORE VISUALIZATION RESULTS OF ABLATION STUDY

We present more visualization results of ablation study in Figure 7 and Figure 8. In Figure 7, The results of the same item across different images are obtained by the same passwords. For instance, "enc1" across "img1" and "img2" are encrypted by the same passwords. "enc1" and "enc2" denote the encrypted results using two distinct passwords. "dec1" and "dec2" denote the decrypted results

**Algorithm 1** Robust Multi-targeted Adversarial Attack

**Input:** $\mathcal{I}_{ori}$, $\mathcal{E}_{tar\_i}$, $i = \{0, 1, .., m\text{-}1\}$
**Output:** $\mathcal{I}_{pro}$
1: $\delta = 0$, $iter = 0$, $\mathcal{I}_{pro} = \mathcal{I}_{ori}$
2: **while** **CosSim**$(\mathcal{E}_{pro\_i}, \mathcal{E}_{tar\_i}) > ths$ **do**
3:    $\mathcal{I}_{pro} = \mathcal{I}_{ori} + \delta$
4:    $\mathcal{I}_{pro} = $ **diff_distortion**$(\mathcal{I}_{pro})$
5:    **for** $i \in [0, m\text{-}1]$ **do**
6:       $\mathcal{E}_{pro\_i} = $ **IE**$_i(\mathcal{I}_{pro})$
7:    **end for**
8:    $\mathcal{L}_{mt} = \sum_{i=0}^{n-1} 1 - $ **CosSim**$(\mathcal{E}_{pro\_i}, \mathcal{E}_{tar\_i})$
9:    $\delta = \delta - \sigma * \nabla_{\mathcal{I}_{adv}} \mathcal{L}_{mt}$
10:    $\delta = \text{clip}(\delta, \text{min}=-\epsilon, \text{max}=+\epsilon)$
11: **end while**
12: **return** Outputs

using two distinct and random passwords. As shown in Figure 7 (b), the loss function $\mathcal{L}_{div}$ enhances the diversity of encrypted and decrypted results using different passwords ("enc1" vs "enc2", "dec1" vs "dec2"). The loss function $\mathcal{L}_{div\_s}$ improves the diversity of encrypted and decrypted results using the same password for different images ("enc1"/"enc2"/"dec1"/"dec2" across "img1" and "img2").

As shown in Figure 8, the loss function $\mathcal{L}_{mt}$ enhances the universality across various zero-shot image-to-image generation methods.

### A.4 MORE VISUALIZATION RESULTS OF ENCRYPTION AND DECRYPTION

We present more visualization results of encryption and decryption in Figure 9 and Figure 10.

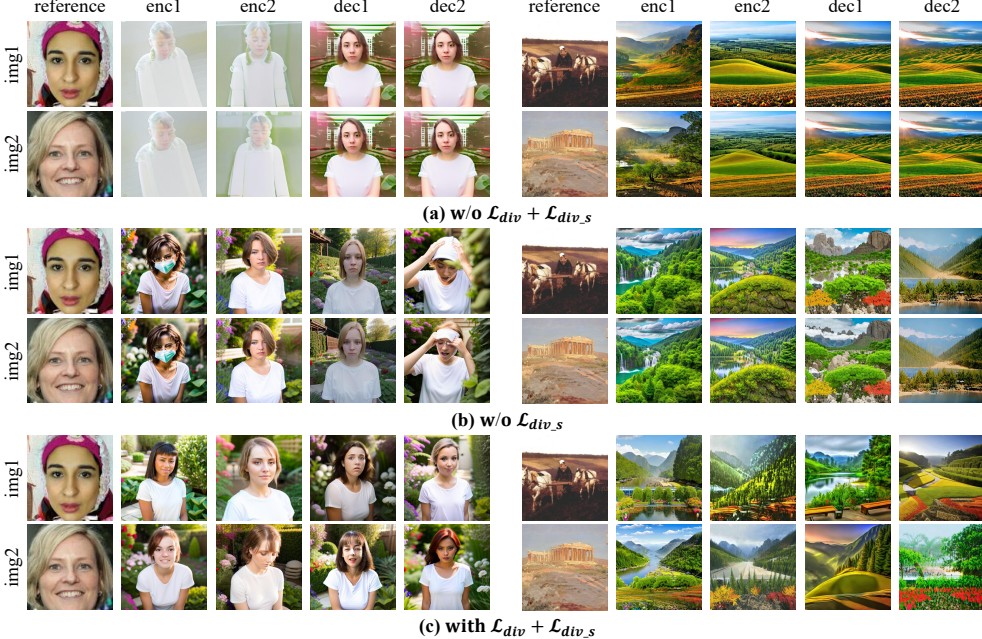

Figure 7: More visualization results of ablation study on loss functions of encryption and decryption. The results of the same item across different images are obtained by the same passwords. For instance, "enc1" across "img1" and "img2" are encrypted by the same passwords. "enc1" and "enc2" denote the encrypted results using two distinct passwords. "dec1" and "dec2" denote the decrypted results using two distinct and random passwords.

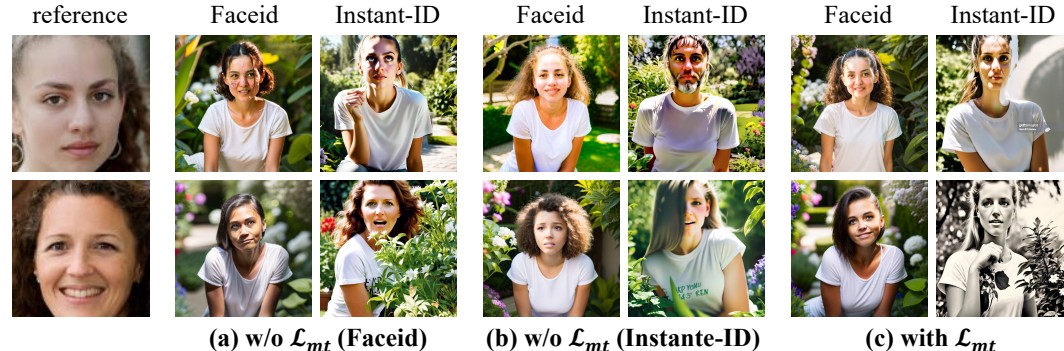

| reference | Faceid | Instant-ID | Faceid | Instant-ID | Faceid | Instant-ID |

**(a) w/o $\mathcal{L}_{mt}$ (Faceid)**   **(b) w/o $\mathcal{L}_{mt}$ (Instante-ID)**   **(c) with $\mathcal{L}_{mt}$**

Figure 8: More visualization results of ablation study on multi-targeted loss function. (a) Minimizes cosine similarity only in the Faceid embedding domain. (b) Minimizes cosine similarity only in the Instant-ID embedding domain. (c) Minimizes cosine similarity in both Faceid and Instant-ID embedding domains.

## A.5 ETHICS STATEMENT

This work adheres to the ICLR Code of Ethics. In this study, no human subjects or animal experimentation was involved. All datasets used, including CelebA, FFHQ, and Wikiart, were sourced in compliance with relevant usage guidelines, ensuring no violation of privacy. We have taken care to avoid any biases or discriminatory outcomes in our research process. No personally identifiable information was used, and no experiments were conducted that could raise privacy or security concerns. We are committed to maintaining transparency and integrity throughout the research process.

## A.6 REPRODUCIBILITY STATEMENT

We have made every effort to ensure that the results presented in this paper are reproducible. The experimental setup, including training steps, model configurations, and hardware details, is described in detail in the paper. Additionally, the datasets used in this paper, such as CelebA, FFHQ, and Wikiart, are publicly available, ensuring consistent and reproducible evaluation results. We believe these measures will enable other researchers to reproduce our work and further advance the field.

## A.7 STATEMENT OF THE USE OF LARGE LANGUAGE MODELS (LLMs)

Large Language Models (LLMs) were used to aid in the writing and polishing of the manuscript. Specifically, we used an LLM to assist in refining the language, improving readability, and ensuring clarity in various sections of the paper. The model helped with tasks such as sentence rephrasing, grammar checking, and enhancing the overall flow of the text.

It is important to note that the LLM was not involved in the ideation, research methodology, or experimental design. All research concepts, ideas, and analyses were developed and conducted by the authors. The contributions of the LLM were solely focused on improving the linguistic quality of the paper, with no involvement in the scientific content or data analysis.

The authors take full responsibility for the content of the manuscript, including any text generated or polished by the LLM. We have ensured that the LLM-generated text adheres to ethical guidelines and does not contribute to plagiarism or scientific misconduct.

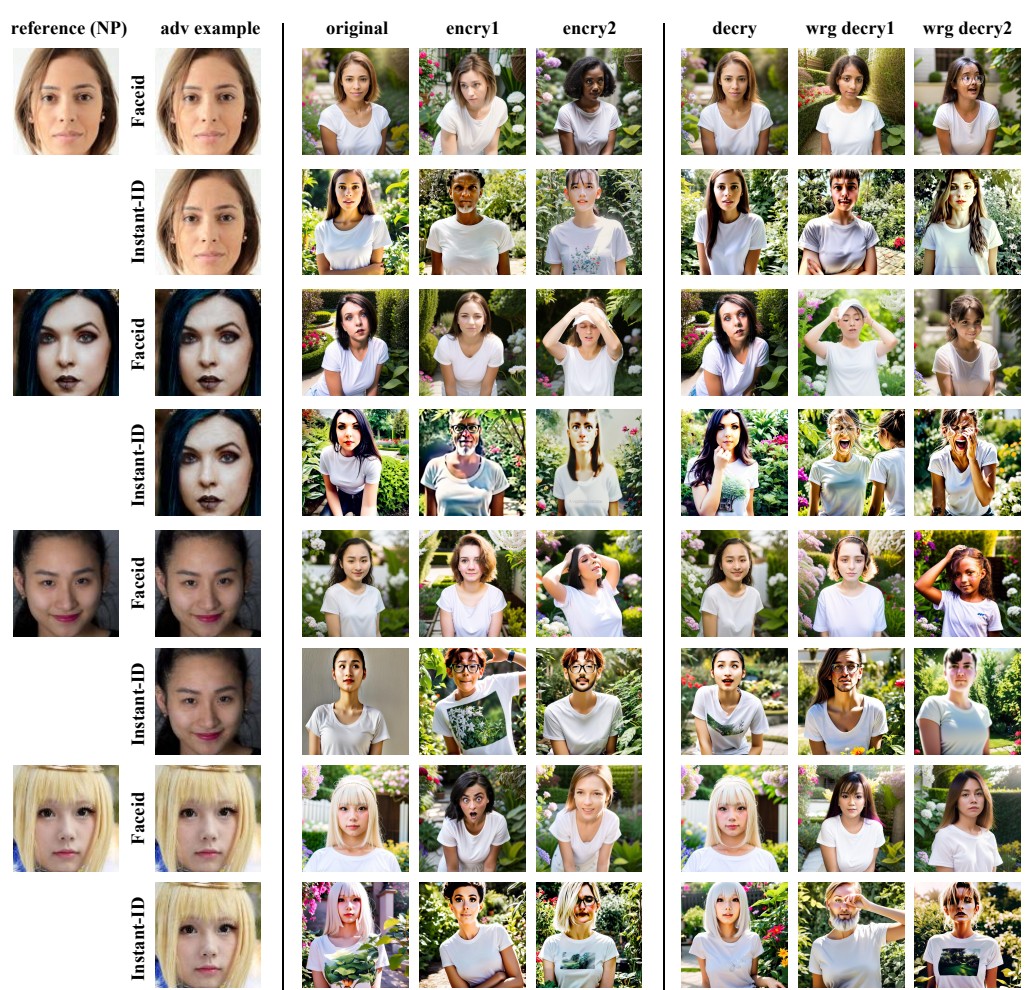

Figure 9: More visualization results of encryption and decryption performance. Text prompts: "*a young woman in white T-shirt in a garden*".

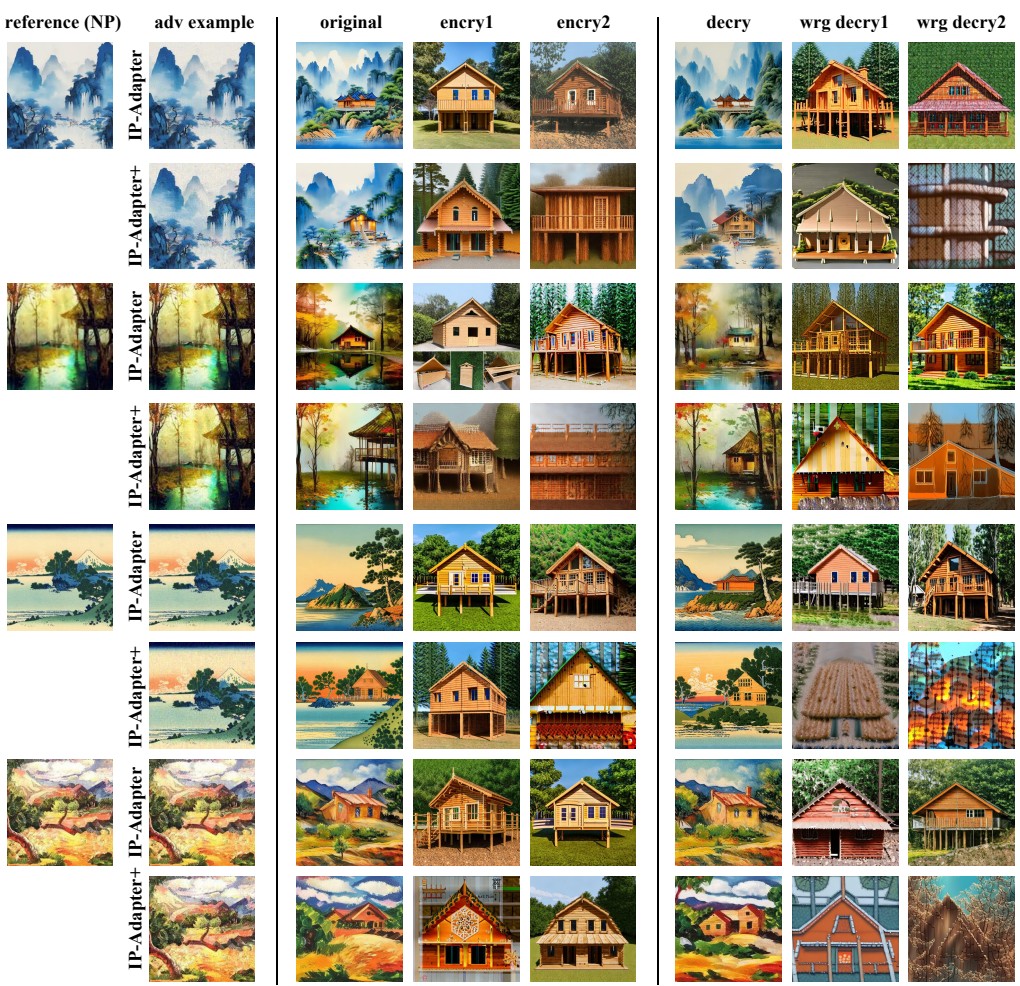

Figure 10: More visualization results of encryption and decryption performance. Text prompts: "*best quality, high quality, a wooden house in forest*".

