# OpenReview forum: "Anti-Adapter Armor: A Universal and Authentication-Integrated Framework for Preventing Unauthorized Zero-Shot Image-to-Image Generation"
_ICLR.cc/2026/Conference — ICLR 2026 Conference Withdrawn Submission_

### Official Review · Reviewer_ED7f · 2025-10-15

**Soundness:** 3
**Presentation:** 3
**Contribution:** 2
**Rating:** 2
**Confidence:** 4

**Summary:**

The paper introduces Anti-Adapter Armor, a purported universal, authentication-integrated framework to protect images against unauthorized zero-shot image-to-image (I2I) generation. It combines embedding encryption with adversarial perturbations (“protective coating”). Experiments are conducted on facial identity protection and artwork anti-plagiarism tasks, claiming superiority over existing baselines.

**Strengths:**

1. Addresses an important and timely problem of protecting personal images from unauthorized zero-shot I2I generation.
2. Proposes a practical framework combining embedding encryption with adversarial perturbations.
3. Provides initial experimental validation on both facial identity protection and style plagiarism tasks.

**Weaknesses:**

**1. Limited Novelty**

* The paper combines adversarial perturbations with password-conditioned embedding transformations. Both ideas are established in the literature, and while the integration is interesting, it may not represent a substantial conceptual advance.
* The claim of being the “first universal and authentication-integrated framework” seems somewhat overstated. Prior work on adversarial cloaking, de-identification, and watermarking has explored similar directions. The contribution, while practical, feels incremental.

**2. Technical Soundness and Clarity**

* The encryption/decryption modules are described at a high level, but the design choice (attention-based architecture) is not fully justified. It would help to clarify why this approach is preferable to simpler baselines (e.g., projection + password mixing).
* The loss formulation appears to be a combination of multiple cosine-similarity terms with heuristic weights. More explanation or theoretical motivation for how these terms interact would strengthen the work.
* The security-related claims (e.g., low wrong decryption rate, password robustness) may give the impression of cryptographic guarantees, which the current framework does not provide. Clarifying the security assumptions and limitations would improve credibility.

**3. Experimental Limitations**

* Experiments are performed on relatively small subsets of CelebA, FFHQ, and WikiArt. The test sets (200 faces, 50 paintings) make it difficult to draw strong conclusions about universality.
* Only two I2I methods per task are tested. Given the diversity of zero-shot generation approaches (ControlNet, fine-tuned diffusion, hybrid encoders), additional coverage would make the evaluation more convincing.
* The evaluation relies on cosine similarity, AFR, and PSNR, but omits user studies or perceptual evaluations. Moreover, robustness to adaptive adversaries and purification strategies is not examined, which is important for a protection framework.

**4. Missing Threat Model**

* A clear definition of the threat model is essential but absent. It is not specified whether attackers are assumed to have white-box, black-box, or gray-box access, nor what their goals and resources are. Without this, the scope of protection is unclear.

**5. Suggested Additional Experiments**

* **Adaptive white-box attacks**: Direct optimization against the encryption/perturbation objective.
* **Black-box query-based attacks**: Gradient-free or query-efficient methods to approximate embeddings.
* **Purification-based attacks**: Denoising, diffusion-based reconstruction, or JPEG recompression.
* **Cross-encoder transfer**: Testing recovery using unseen encoders to assess generalization of protection.

**Questions:**

W1, W2, W4

---

### Official Review · Reviewer_BHeY · 2025-10-16

**Soundness:** 3
**Presentation:** 3
**Contribution:** 2
**Rating:** 4
**Confidence:** 4

**Summary:**

This paper introduces a framework called "Anti-Adapter Armor," which aims to prevent unauthorized face forgery or art style mimicry by AI models by adding a password-protected adversarial "coating" to images. The core mechanism is that this coating misleads the AI model into extracting an encrypted, incorrect image feature, thus generating distorted content. Authorized users, however, can use the correct password with a decryptor to recover the correct feature for normal use.

The idea behind this work is very interesting. However, its core contribution is more a clever integration of existing technologies than a fundamental methodological or theoretical breakthrough. This makes it a less suitable fit for ICLR.

**Strengths:**

- Importance and Practical Value of the Problem: This paper addresses a critical and pressing real-world issue in the era of AI-generated content (AIGC)—how to protect individuals' portrait rights and artists' intellectual property. This represents a research direction with significant practical value.
- Comprehensive Framework Design: The proposed framework demonstrates thorough consideration, with its integrated “authentication mechanism” standing out as a key highlight. Compared to previous irreversible protection methods, cryptography-based authorized usage significantly enhances the scheme's flexibility and practicality.
- Thorough Experimental Evaluation: The authors conducted comprehensive experiments covering two major scenarios—facial identity protection and artistic style anti-plagiarism—and compared their framework with multiple existing methods. This validated the framework's effectiveness and versatility.

**Weaknesses:**

- Limited Novelty for ICLR: The primary weakness is the work's limited methodological novelty for a venue like ICLR. While the framework is a clever and valuable application, it primarily combines existing techniques (adversarial attacks, attention networks) rather than introducing a fundamental algorithmic breakthrough in machine learning.

- Lossy Encryption Mechanism: The proposed "encryption-decryption" cycle is inherently lossy, as its objective is to maximize cosine similarity rather than achieve perfect, bit-for-bit reconstruction. This functional reversibility, while practical, differs fundamentally from true cryptographic invertibility and may not be suitable for all use cases.

- Limited Robustness: The framework's robustness against common image transformations is a significant concern. The authors explicitly acknowledge its vulnerability to JPEG compression, a ubiquitous format online. Furthermore, the paper lacks evaluation against more sophisticated adversarial purification techniques, leaving its resilience in a practical threat landscape an open question.

**Questions:**

- Could the authors clarify what constitutes the core algorithmic innovation, beyond the integration of established components?

- The proposed “encryption–decryption” cycle appears inherently lossy, as it aims to maximize cosine similarity rather than achieve exact reconstruction. How do the authors reconcile this with the notion of encryption, which typically implies perfect invertibility?

- The authors acknowledge vulnerability to JPEG compression—a common transformation in online environments. How might the proposed framework be adapted to handle such lossy compression while preserving privacy effectiveness?

---

### Official Review · Reviewer_tyLK · 2025-10-25

**Soundness:** 2
**Presentation:** 1
**Contribution:** 3
**Rating:** 4
**Confidence:** 4

**Summary:**

The paper introduces Anti-Adapter Armor, a framework that adds imperceptible perturbations to protects images against ID-preserving generation or image plagiarism. At the center of the method is a multi-target adversarial attack. The protection is also associated with a password, which could be used by authorized users to decrypt the image embedding useful for generation. Experiment results demonstrated the effectiveness of this method in protection and decryption, robustness against certain distortions, and ablations about the encryption/decryption details.

**Strengths:**

-	The method protects the content while still allowing authorized users to bypass the protection with a specific key for legitimate regeneration. This is a novel and practical design.
-	Compared to existing methods, the protection maintains good visual quality, introducing fewer visible artifacts.
-	Although not explicitly demonstrated by the authors, the method should have a significantly faster image processing speed than optimization-based methods.

**Weaknesses:**

-	The writing in Section 4 (Experiments) is largely descriptive and lacks discussion. The authors present results in Tables and Figures, but do not adequately interpret them to draw clear conclusions or provide justifications. The paper would be significantly strengthened if the authors explicitly connect the results presented to the properties/advantages of the proposed method.
-	The robustness evaluation is insufficient. The experiments cover common image processing distortions like noise and blur, but do not explore robustness to pixel-misaligned scenarios, such as rotation or scaling, so it remains unclear whether the protection would still be effective under these common transformations.
-	The experiments are confined to InstantID and a few variants of IP-Adapter. It is unclear whether the protection would still be effective against unseen regeneration models, such as Midjourney.
-	Several factual errors:
  - Line 77: IDProtector is not a data-poisoning method, but an adversarial attack-based method, which functions similarly to the approach in this work. Methods like Anti-Dreambooth/SimAC are data-poisoning methods.
  - Line 81: IDProtector is an open-source method.
  - Line 86: It is not true that protective methods are generally vulnerable to image post-processing. On the contrary, many methods (such as IDProtector) have demonstrated considerable robustness to such techniques.

**Questions:**

-	Why artwork similarity is measured using ESM? How reliable is this metric?
-	The epsilon budget for artwork protection (21/255) is very large, yet the perceptual distortion shown in the images appear minimal. What could be the reasons for this apparent inconsistency?
-	Why were different similarity thresholds and epsilon budgets used for the facial identity protection and the artwork anti-plagiarism settings? Would it be possible for one image to simultaneously achieve both protections while maintaining the currently reported performance?
-	What are the epsilon budgets used for the baseline methods?

---

### Official Review · Reviewer_utEG · 2025-11-01

**Soundness:** 2
**Presentation:** 3
**Contribution:** 3
**Rating:** 4
**Confidence:** 4

**Summary:**

This paper proposes Anti-Adapter Armor, a proactive protection framework against unauthorized zero-shot image-to-image generation. The method introduces (1) a reversible embedding encryption mechanism conditioned on a password, and (2) a multi-target adversarial protective coating that perturbs images. It blocks unauthorized users while allowing authorized users to use the encrypted images. Experiments show substantial protection efficacy and robustness to common distortions.

**Strengths:**

1. The paper proposes a novel framework for protecting images from unauthorized zero-shot image-to-image generation while providing an authentication mechanism that allows authorized users to generate proper images.
2. The paper writing is good and easy to follow.
3. Experimental results show better protection performance compared to baselines.
4. Various losses are designed to properly encrypt the image embedding.

**Weaknesses:**

1. It seems the proposed method requires using the same image encoder as the users. What if the users use an unknown image encoder that is not considered during optimization?
2. In Tab 3, it will be interesting to see the evaluation under more sophisticated adversarial purification methods such as DiffPure (https://arxiv.org/abs/2205.07460) in addition to simple image distortions. In addition, the results can be compared with baselines to understand the relative robustness of the proposed method.
3. While the authors claim that authorized users can use the image for image generation, the experiment results are mainly on preventing unauthorized users, and there is no quantitative result on the image generation performance when the correct password is used for decryption.
4. Regarding the motivation of the work, can "Bob" just provide "Alice" (the authorized user) the original unprotected image without the need for the authorization mechanism? What are the benefits of the current approach? This should be discussed in the paper.

**Questions:**

See the weakness section.

---

### Note · Authors · 2025-11-12

I have read and agree with the venue's withdrawal policy on behalf of myself and my co-authors.